# Hyperstretching DNA

Koen Schakenraad[1,2], Andreas S. Biebricher[3], Maarten Sebregts[1], Brian ten Bensel[3], Erwin J.G. Peterman [3]
Gijs J.L. Wuite[3], Iddo Heller [3], Cornelis Storm[1,4] & Paul van der Schoot[1,5]

The three-dimensional structure of DNA is highly susceptible to changes by mechanical and biochemical cues in vivo and in vitro. In particular, large increases in base pair spacing compared to regular B-DNA are effected by mechanical (over)stretching and by intercalation of compounds that are widely used in biophysical/chemical assays and drug treatments. We present single-molecule experiments and a three-state statistical mechanical model that provide a quantitative understanding of the interplay between B-DNA, overstretched DNA and intercalated DNA. The predictions of this model include a hitherto unconfirmed hyperstretched state, twice the length of B-DNA. Our force-fluorescence experiments confirm this hyperstretched state and reveal its sequence dependence. These results pin down the physical principles that govern DNA mechanics under the influence of tension and biochemical reactions. A predictive understanding of the possibilities and limitations of DNA extension can guide refined exploitation of DNA in, e.g., programmable soft materials and DNA origami applications.

[1] Department of Applied Physics, Eindhoven University of Technology, P.O. Box 5135600 MB Eindhoven, The Netherlands. [2] Instituut-Lorentz and Mathematical Institute, Universiteit Leiden, P.O. Box 95062300 RA Leiden, The Netherlands. [3] Department of Physics and Astronomy, LaserLaB Amsterdam, Vrije Universiteit Amsterdam, De Boelelaan 1081, 1081 HV Amsterdam, The Netherlands. [4] Institute for Complex Molecular Systems, Eindhoven University of Technology, Eindhoven, 5612 AJ, The Netherlands. [5] Institute for Theoretical Physics, Utrecht University, Princetonplein 5, 3584 CC Utrecht, The Netherlands. Koen Schakenraad, Andreas S. Biebricher contributed equally to this work. Iddo Heller, Cornelis Storm and Paul van der Schoot jointly supervised this work. Correspondence and requests for materials should be addressed to K.S. (email: schakenraad@lorentz.leidenuniv.nl) or to A.S.B. (email: a.s.biebricher@vu.nl)

The elastic properties of double-stranded (ds) DNA play a crucial role in many cellular processes. Indeed, replication, transcription and the histone-mediated compaction into chromatin all involve significant deformation of the stiff polynucleotide chain. The elastic properties of dsDNA are intimately linked to the characteristic helical structure of the molecule[1]. The interplay between DNA mechanics and structure is particularly evident in stretching experiments[2,3]. While the bond lengths in the backbone of DNA permit a maximum extension of 0.7 nm per base pair, the native helical B-form of DNA has a length of only 0.34 nm per bp[4]. This corresponds to the dense stacking of the planar aromatic structures of the bases and resembles the spacing of, e.g., stacked layers of graphene. At a critical force of about 65 pN, dsDNA undergoes a cooperative overstretching transition in which the helix partly unwinds. This results in a sudden 70% increase of the extension to 0.58 nm per bp. This overstretched state can consist of base-paired DNA (such as S-DNA), non-base-paired DNA or coexisting base-paired and non-base-paired states, depending on subtle differences in ionic strength, temperature and the local nucleotide sequence[5,6]. That 0.58 nm per bp is significantly shorter than the naive limit of 0.7 nm per bp, set by the backbone of DNA, has been attributed to a more compact bond configuration in the sugar groups of overstretched DNA[3].

Fluorescent probes are often used to shed light on local and global structural changes in dsDNA, caused in vitro by stretching and/or twisting and in vivo, e.g., by the action of proteins during transcription[7–9]. Many DNA-binding probes are intercalators, molecules with planar moieties that insert themselves between adjacent base pairs[10,11], thereby gaining a binding free energy. Dense stacking interactions of the aromatic systems of typical planar mono-intercalators such as YO-PRO-1 and Sytox Orange and bis-intercalators such as YOYO-1 and POPO with neighbouring base pairs result in an extension of DNA by 0.34 nm per intercalated moiety[12–14]. Because binding of such intercalators locally extends DNA by 100% to 0.68 nm per bp, intercalation is not commensurate with the natural local geometry of the dsDNA. This implies a significant perturbation of the double helix, which is leveraged, e.g., in the context of rational design of drugs aimed to disrupt the physiological functioning of DNA[15,16].

Despite the local doubling of DNA extension, an intercalated DNA molecule is typically at most 50% elongated, because next-neighbour exclusion in principle permits only every second base pair to be intercalated[17]. Observations made at very high intercalator concentration[18], however, suggest that next-neighbour exclusion can be overcome. In this contribution, we demonstrate that this is indeed the case due to the combined effect of mechanical and mass action.

A comprehensive, quantitative understanding of the physical interplay between B-DNA, overstretched DNA and intercalated DNA, and the effects of tension and intercalator concentration, is currently lacking. This leads us to our central research question: 'What aspects of the mechanics of dsDNA are affected by intercalative binding, and to what extent, in particular for stretching forces near and beyond the overstretching transition?'. To address this question, we combine force-fluorescence microscopy, in which we mechanically stretch individual dsDNA molecules, while visualizing the binding of the fluorescent intercalator YO-PRO-1, with predictions of a statistical mechanical model, and provide a quantitative understanding of the interplay between B-DNA, overstretched DNA and intercalated DNA.

As we shall see, high-force DNA mechanics is governed by a competition between overstretching and the action of intercalators. At high stretching forces, this may, in the presence of intercalators, lead to a cooperative transition from the overstretched to a novel hyperstretched (HS) and fully intercalated state of twice the length of B-DNA.

## Results

We quantified the force-extension behaviour of DNA by stretching 16.4 μm long phage lambda DNA strands between two optically trapped microspheres in aqueous solution containing a wide range of concentrations of the intercalator YO-PRO-1. In contrast to previous work[18], experiments were conducted in a buffer that favours DNA overstretching without the occurrence of single-stranded DNA sections originating from nicks and ends[19–22] ('peeling', see Methods and Supplementary Fig. 1). We furthermore choose a range of DNA extension rates (30–500 nm s⁻¹) that are sufficiently low for intercalator binding and unbinding to be in equilibrium at each point of the force-extension curve (Supplementary Fig. 2). In Fig. 1, we plot our findings.

The striking effect of intercalative binding on the force-extension curves is obvious from Fig. 1. There are three key features, as indicated in Fig. 1: (1) DNA intercalation results in extension of the DNA beyond its bare contour length of 0.34 nm per bp at forces below the overstretching transition at 65 pN; (2) a shift of the overstretching transition to higher forces with increasing YO-PRO concentration up to 1 μM; (3) the overstretching transition vanishes at YO-PRO concentrations above 1 μM. At these concentrations, the DNA extends beyond 0.58 nm per bp in the high-force limit. Explicitly excluding the possibility of peeling, this suggests that next-neighbour exclusion of intercalators can indeed be overcome.

Feature 1 is intuitive because mass action and DNA tension force more intercalators to insert themselves between the base pairs and lengthen the DNA, as has been quantified before[14]. Feature 2, however, is not that intuitive: more force is required for overstretching despite the lengthening effect of intercalation. To rationalize this feature, as well as feature 3, we construct a simple three-state statistical mechanical model that we solve analytically (Methods). Each base pair is either in the B-DNA state, the overstretched state, or the intercalated state, accounting for the appropriate length and bending and stretching stiffness contributions to the state of the chain. All the base pairs are presumed to be in thermodynamic equilibrium with each other and with a reservoir containing the intercalators. Our model incorporates ingredients from earlier works[23–27], but is the first to combine intercalation and overstretching in the model.

Our model-based explanation of feature 2 hinges on a free energy penalty that we associate with neighbouring overstretched and intercalated base pair sequences. This causes the intercalators to unbind in order for the overstretching transition to take place, as in fact was previously observed by Biebricher et al[14]. Because this happens at the expense of the binding energy, this must be compensated for by a larger overstretching force. The value of

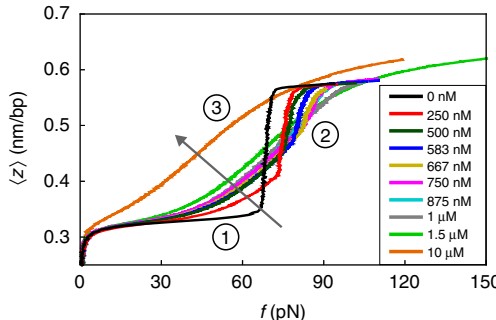

**Fig. 1** The extension z per base pair as a function of the stretching force f for single dsDNA molecules in aqueous solution containing a wide range of concentrations of the intercalator YO-PRO. The arrow indicates the direction of increasing concentration. The key features described in the main text are indicated by 1, 2 and 3

this free energy penalty that we establish for YO-PRO is equal to 3.5 times the thermal energy $k_B T$, equivalent to 8.7 kJ mol$^{-1}$ at room temperature (see Methods and Supplementary Note 1). As usual, $k_B$ denotes Boltzmann's constant and $T$ the absolute temperature.

Feature 3 can be explained by presuming that next-neighbour exclusion of intercalators is not absolute but comes at a finite free energy cost. For YO-PRO, we find 1.5 $k_B T$ (see Methods and Supplementary Note 1). Since this free energy cost is finite, next-neighbour exclusion can be overcome by a combination of mechanical and chemical work, i.e., by increased tension and mass action. This results in the emergence of a novel HS state above a critical concentration of 1 µM YO-PRO, where the DNA adopts a conformation that has twice the native dsDNA length per base pair and double the intercalator coverage (compared to the most densely covered staggered intercalation state).

We point out that the three features are also observed in previously published force-extension experiments on dsDNA in the presence of the intercalator ethidium bromide (EtBr)[18,28], indicating that they are not specific to YO-PRO, but are in fact representative of a larger group of intercalators.

Our theoretical force-extension curves, obtained from a fitting procedure to the data in Fig. 1 (see Methods and Supplementary Note 1), are given in Fig. 2a, and accurately describe all three features characterizing our experimental findings of Fig. 1 for all forces and concentrations probed, as well as the cooperative nature of the bare overstretching transition[26,27]. Additionally, the figure suggests the existence of HS-DNA below the critical concentration of 1 µM via a novel transition from the overstretched to the HS state. The sharpness of this hyperstretching transition, indicated by H in Fig. 2, is governed by whether next-neighbour intercalation is more or less penalized than neighbouring overstretched and intercalated base pairs (Supplementary Note 1). According to our model, all force-extension curves should ultimately attain the HS-DNA state in the presence of intercalators.

It is important to stress that our fitted model parameters are kept constant for the various concentrations of YO-PRO. The full details of the theory as well as our fitting procedure and the values of our fitting parameters can be found in the Methods and Supplementary Notes 1–3. Fitting our model to the experimental force-extension curves produces a prediction for the fraction of intercalated base pairs as a function of force and concentration given in Fig. 2b, amenable to experimental verification. Our explanation for features 1, 2 and 3, given above, are confirmed by this figure.

A crucial prediction, highlighted by H in Fig. 2b, is that the hyperstretching transition is characterized by reintercalation of YO-PRO leading to complete intercalation of the DNA. This, and also the alternative route to the HS state (feature 3), we test experimentally. For this, we invoke a wide-field epifluorescence microscopy layout to directly detect the presence of YO-PRO on optically stretched DNA (Methods)[14]. We adopt stretching and illumination conditions that minimize the photodamaging properties of the intercalating dye to the DNA molecule[13], and exploit the 2–3 orders of magnitude enhancement in fluorescence of YO-PRO upon DNA intercalation.

Complementary to our force analysis, the fluorescence measurements in fact provide spatially resolved information on intercalation and allow for detecting any undesired peeling (Supplementary Fig. 3). More importantly, the fluorescence signal is an unequivocal proof of intercalator presence, as opposed to lengthening observed in force-extension curves that can also be caused by peeling (Supplementary Fig. 1). Figure 3a, b show two kymographs that display DNA stretching experiments in the presence of 875 and 1500 nM YO-PRO. The total fluorescence intensities, which are a measure for the number of bound

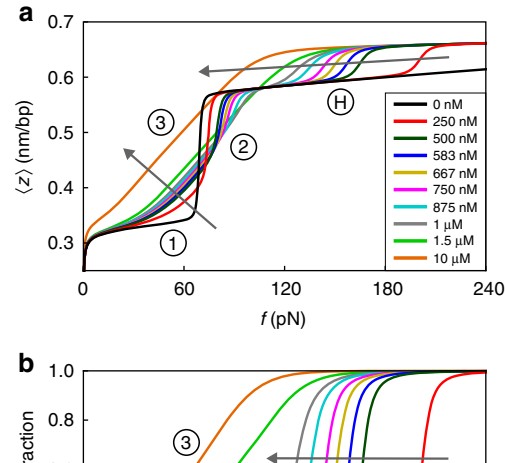

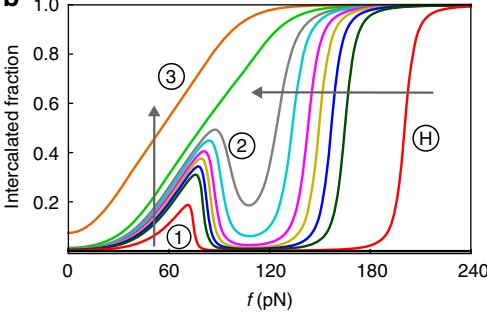

**Fig. 2** Theoretical curves of the DNA extension per base pair (**a**) and fraction of intercalator-bound base pairs (**b**) as a function of the applied force for intercalator concentrations $C$ that match the experimental ones of Fig. 1. The arrows indicate the direction of increasing concentration. The model parameters, obtained from a fitting procedure, are discussed in Supplementary Note 1. The features 1, 2 and 3 and the prediction H are discussed in the main text

intercalators, are plotted as a function of the stretching force in Fig. 3c for both concentrations (Methods). In order to minimize DNA photodamage, a lower illumination intensity was used for the higher intercalator concentration.

The similarities between Fig. 3c (experiments) and Fig. 2b (theory) are striking. Focusing first on the data obtained at 875 nM (Fig. 3a), i.e., at a concentration that retains the overstretching transition (Fig. 1), we notice a gradual brightening for forces up to the overstretching transition, corresponding to region 1 in Fig. 2b. Subsequently, dark regions appear in the kymograph, signifying the transition to S-DNA that causes the intercalators to detach from the DNA, as explained above. This corresponds to region 2 in Fig. 2b.

This darkening translates into a drop of the fluorescence intensity in Fig. 3c (blue circles) and confirms that overstretching and intercalation are mutually exclusive. Importantly, for even larger forces, bright regions start reappearing with even higher intensities than before the overstretching transition, corresponding to region H in Fig. 2b. The final intensity is approximately double that of the intensity prior to the overstretching transition. This constitutes the first unambiguous proof that at a critical force beyond the overstretching transition, intercalators rebind to fully intercalate the DNA, producing the predicted HS-DNA state.

At the higher intercalator concentration of 1500 nM, on the other hand, we observe a gradual brightening of fluorescence with no apparent darkening zone in between (Fig. 3b, c, green squares), corresponding to region 3 in Fig. 2b (theoretical data). Indeed, in the 1500 nM force-extension curve (Figs. 1 and 2a, comprising experimental and theoretical data, respectively), there is no indication of an overstretching transition (feature 3). This demonstrates that the lengthening in the force-extension curves

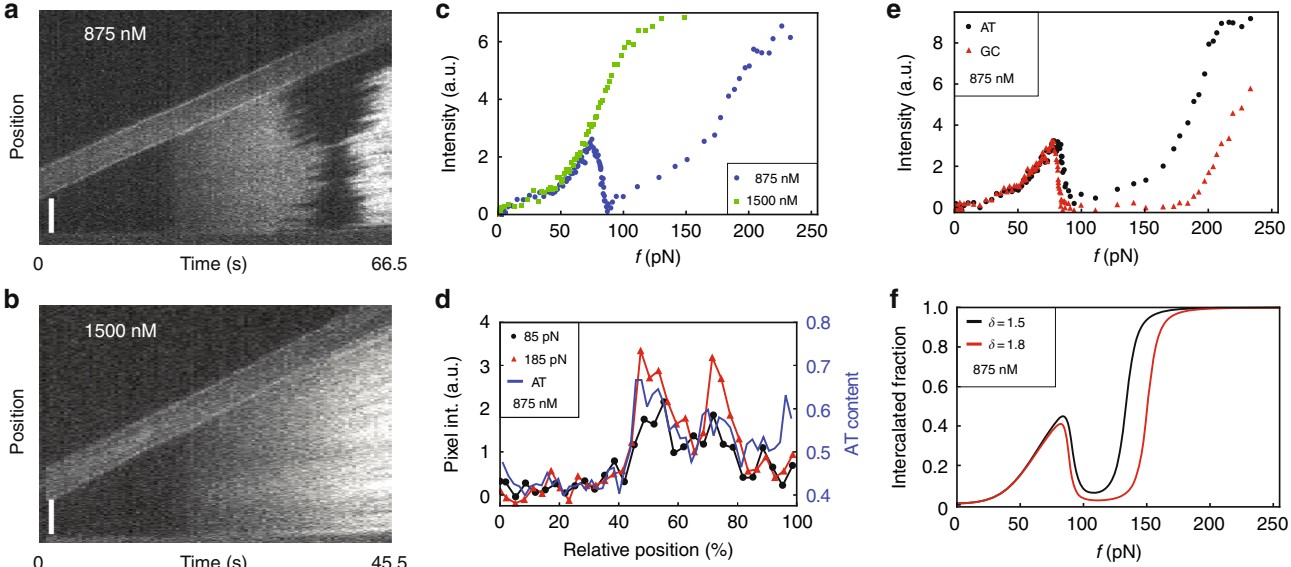

**Fig. 3** Force-fluorescence experiments confirm the existence of hyperstretched (HS-) DNA. **a, b** Kymographs of stretching experiments in the presence of 875 and 1500 nM of the intercalator YO-PRO (length of the scale bar is 5 μm; here the bright ~5 μm wide band shows one bead moving upwards to stretch the DNA). **c** Integrated fluorescence intensities as a function of force for 875 nM YO-PRO (blue circles) and 1500 nM YO-PRO (green squares). Intensities in different kymographs have been scaled to facilitate comparison. **d** The local fluorescence intensity at stretching forces 85 and 185 pN and 875 nM YO-PRO concentration is strongly correlated with the local AT content of the DNA. **e** Intensity as a function of force for AT- and GC-dominated regions at 875 nM YO-PRO. **f** A difference in the free energy penalty for neighbouring intercalated base pairs δ qualitatively explains the difference between AT- and GC-dominated regions seen in **e**. For discussion, we refer to the main text

beyond 0.58 nm per bp can only be explained by the presence of HS-DNA and not by potential peeling[29].

An interesting feature of the fluorescence data is the inhomogeneous distribution of the fluorescence intensity in the unbinding and rebinding processes. We find that this darkening/brightening pattern in the fluorescence kymographs is not only highly reproducible, but also independent of intercalator concentration and the used dye (Supplementary Fig. 4). Figure 3a suggests the existence of domains, characterized by different critical forces of unbinding and rebinding. Indeed, as Fig. 3d demonstrates, there is a clear correlation between the fluorescence intensity at 85 and 185 pN stretching forces and 875 nM YO-PRO concentration and the local AT content. Closer inspection of the fluorescence pattern in the kymographs and comparison with the sequence of the lambda DNA supports this hypothesis (Supplementary Fig. 5). These two forces highlight the difference between AT-rich and GC-rich regions, seen also in Fig. 3a by a wider or less wide dark band in the kymograph, where there is very little intercalation. The two forces of 85 and 185 pN fall within the non-intercalated force range of the upper GC-rich region but outside the non-intercalated force range of the lower AT-rich region in the kymograph. We separately integrate the fluorescence intensity over the GC-poor and GC-rich regions of phage lambda DNA and show the results in Fig. 3e, confirming that AT-rich DNA exhibits a narrower non-intercalated force-extension range than does GC-rich DNA.

Within our theoretical model, this difference in behaviour between GC-rich and GC-poor regions can be captured by varying a single parameter, being the free energy penalty for two neighbouring intercalated base pairs, i.e., the extent of next-neighbour exclusion. Figure 3f shows the theoretical fraction of intercalated DNA segments for values of this free energy penalty of 1.5 $k_BT$ (black) and 1.8 $k_BT$ (red), keeping all other parameters of our model the same as those used in Fig. 2. The shapes of the theoretical curves show remarkable agreement with the data in Fig. 3e albeit that theory and experiment disagree about the

critical reintercalation force. Below we link this observation to differences in so-called gamma torsion[30] for the two types of complementary base pair.

Another remarkable prediction of our model pertains to feature 2. According to our model calculations (Supplementary Note 3), the theoretical overstretching force is, for all intents and purposes, a linear function of the concentration $C$,

$$f_{os}(C) \approx f_{os}(0) + \frac{k_BTC}{l_0 C_w} \exp\left(\frac{\gamma_2 - 1}{\gamma_1 - 1}\varepsilon_1 - \varepsilon_2\right), \quad (1)$$

where $f_{os}(C)$ is the concentration-dependent overstretching force and $f_{os}(0)$ is the bare B–S transition force of 65 pN. Here, $\varepsilon_1$ and $\varepsilon_2$ are the free energy costs, in units of thermal energy, of converting a single base pair from B-DNA to overstretched and intercalated DNA. $C_w = 55.6$ M is the molar concentration of water ($C/C_w$ is the mole fraction of intercalator in dilute solution), $l_0$ the length of a base pair in the B-DNA state, and $\gamma_1$ and $\gamma_2$ the factors by which overstretched and intercalated DNA are longer than B-DNA. We verify this prediction by analysing our data on the intercalator YO-PRO, and re-analysing previously reported force-extension curves for the intercalator EtBr[18].

The overstretching force shown in Fig. 4 is plotted as a function of the intercalator concentration for both YO-PRO and EtBr, scaled to the concentration required to shift the overstretching force to 75 pN (542 nM for YO-PRO, 6.8 nM for EtBr). As predicted by our model, we find a linear relation between overstretching force and concentration for both intercalators. Not only that, the predicted slope from our fitting to the force-extension curves presented in Fig. 1, and those presented in ref. [18], is quantitatively correct. We refer to Supplementary Note 1 for all the parameter values. We predict a slope of $1.6 \times 10^{-2}$ pN nM$^{-1}$ for YO-PRO, consistent with the observed slope of $(1.7 \pm 0.1) \times 10^{-2}$ pN nM$^{-1}$. For EtBr, we predict 1.0 pN nM$^{-1}$, comparable with the observed slope of $(1.4 \pm 0.1)$ pN nM$^{-1}$. The difference in slope of about two orders of

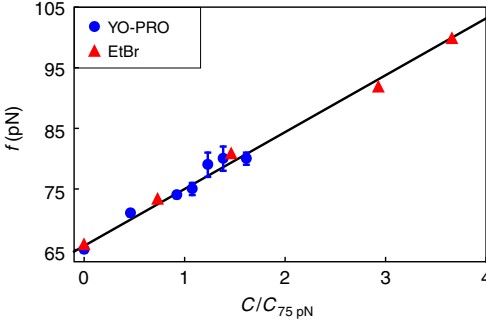

**Fig. 4** Overstretching force as a function of intercalator concentration for YO-PRO (blue circles, error bars, s.d., $\langle n \rangle = 6.3$) and ethidium bromide (red triangles) fitted with a linear relation. The concentrations are scaled on the concentration required to shift the overstretching force to 75 pN (542 nM for YO-PRO, 6.8 nM for EtBr). The obtained slopes are $(1.7 \pm 0.1) \times 10^{-2}$ pN nM$^{-1}$ for YO-PRO and $(1.4 \pm 0.1)$ pN nM$^{-1}$ for EtBr. The data on EtBr is extracted from results previously published in ref. [18]

magnitude between YO-PRO and EtBr is caused by a large difference in the intercalator binding free energy, signifying that EtBr ($\varepsilon_2 = -18 k_B T$) binds more strongly to DNA than YO-PRO does ($\varepsilon_2 = -13 k_B T$).

## Discussion

Our direct observation of HS dsDNA with an extension of ~0.7 nm per bp has several structural implications. In fact, Bustamante and colleagues already noted the suboptimal length of 0.58 nm per bp of overstretched DNA in 1996, and attributed this to a more compact sugar conformation associated with it[3]. However, ssDNA can be stretched up to 0.7 nm per bp[31]. For poly(dA), this occurs already below 150 pN, while for the other poly(dN)s as well as for random sequences, this requires much higher tension[31]. Recent work shows that for ssDNA, the resistance to an extension of 0.7 nm per bp can be attributed to the required flipping of the dihedral sugar angle, referred to as gamma torsion[30].

On account of this, we propose a structural explanation that accounts for all our observations on dsDNA. First, during overstretching, dsDNA only lengthens to 170% of the unperturbed length. We put forward that further lengthening does not occur for forces in the range usually applied experimentally, exactly because of the gamma torsion. For the same reason, >50% intercalation does not easily occur, because this would stretch the backbone towards 200% of the unperturbed length. Further lengthening of the DNA would require overcoming the gamma torsion barrier. Our experiments and modelling suggest that this can be achieved by applying a sufficiently large force in the presence of intercalators. This results in a lengthening of 16%, so from maximum 170 to ~200%, caused by the full intercalation. This we denote as HS-DNA.

The lower gamma torsion barrier for poly(dA) as compared to that for the other poly(dN)s, suggests that AT must have a lower barrier than GC, and hence that AT-rich dsDNA must have a lower hyperstretching force. This is indeed what we observe, see Fig. 3e. In our model, we explain the difference in the hyperstretching transitions of GC- and AT-rich regions by a difference in the free energy penalty associated with neighbouring intercalated base pairs. Hence, if our hypothesis of gamma torsion is correct, then next-neighbour exclusion of intercalators is a direct consequence of this gamma torsion barrier.

Importantly, our fluorescence observations indicate that the HS state consists primarily of intercalator-bound dsDNA, rather than unpeeled ssDNA that does not bind intercalator. Hence, the HS state must be an admissible conformation of the dsDNA itself with intact backbone connectivity. This conformation may be facilitated by intercalators, but we hypothesize that it should be available to naked dsDNA as well, although it might only show up in the extremely high-force regime.

Even though it is not known if this extreme stretching regime is of physiological relevance, DNA is also increasingly used as a component (or even the sole constituent) of soft materials that capitalize on the programmable interactions DNA offers (e.g., DNA gels[32–34] and DNA-coated colloids[35]). For such materials, the HS region can and should feature, and must lead to nonlinear elastic response regimes that previously have not been noticed let alone probed.

We have shown by combining results from force-fluorescence microscopy studies and predictions from a statistical mechanical model that the presence of intercalators in the aqueous solution has a pronounced effect on the mechanical response of dsDNA. Intercalators suppress the overstretched state if present at a sufficiently high concentration, but also give rise to a new, HS form of DNA. At lower concentrations, the overstretching transition shifts toward higher forces but remains observable. In that case, hyperstretched or HS-DNA is obtained through a second transition that takes place far beyond 65 pN. In HS-DNA, next-neighbour exclusion is overcome, leading to a doubling of both the intercalator coverage and the length of the DNA. Our work pins down the physical principles that govern DNA mechanics under the influence of tension and the reversible binding of intercalators. A predictive understanding of the possibilities and limitations of DNA extension, in the presence or absence of intercalator, may guide the refined exploitation of DNA, e.g., in programmable soft materials and DNA origami applications.

## Methods

**Experiments**. All force-fluorescence measurements were conducted on an instrument that combines double optical tweezers with epifluorescence microscopy in a multi-channel flow cell, which has been described earlier[14]. The mono-intercalator YO-PRO-1 Iodide was purchased from Thermo Fisher Scientific and diluted to the desired concentration from a DMSO stock solution. We focused on this particular dye because it is the fluorescent intercalator with the fastest known off-rate[14]; this is crucial in order to achieve intercalation equilibration at the required stretch speeds (see below). In all channels, we used a buffer of 20 mM HEPES, pH 7.5, 1000 mM NaCl and 0.02% Tween 20. The application of high salt in this case is necessary to speed up equilibrium binding of the intercalator[14] and to minimize the formation of single-stranded DNA due to end-peeling[29]. We used a DNA from the λ-phage (48,502 bp = 16.4 μm length), which was modified with several biotins on each end using a standard protocol[36], and caught between two polystyrene beads coated with streptavidin (4.95 μm diameter, Spherotech). The presence of more than one biotin on each end is necessary to allow stretching beyond 200 pN, but precludes the use of a topologically closed DNA construct to reduce peeling[5]. In order to obtain equilibrium binding force-extension curves of DNA in the presence of intercalator, we stretched DNA at different extension/retraction speeds and used the fastest stretching curves that displayed minimal hysteresis (Supplementary Fig. 2). For each condition, we measured between 5 and 10 different DNA molecules, and we recorded at least three stretching curves for each DNA to ensure reproducibility. Force-extension data was selected using a home-written software in LabVIEW (National Instruments) and analysed for display using Origin 9.1 (OriginLab Corporation), fitting to the three-state model was accomplished via a code written in MATLAB (Mathworks). For recording of the fluorescence signal, we used the minimum illumination intensity that still allowed a decent fluorescence signal at medium dye coverage for the used acquisition time (0.5 s). In order to study the transition to HS-DNA, we generally used stretching speeds that were a little bit faster than used for equilibrium binding curves (200–800 nm s$^{-1}$). Fluorescence data were analysed and intensity quantified using the program ImageJ[37]. Data acquisition was usually stopped, and data not considered for analysis, in the case that obvious nicks appeared on the DNA, visible as homogeneously dark patches broadening with increased stretching (Supplementary Fig. 3).

**Theory**. The essence of our model is that each discrete unit of the chain is in one of three states: reference, overstretched or intercalated. This basic principle is most clearly illustrated in the context of freely-jointed chain models. In Supplementary Note 3, we detail how this may be corrected for bending stiffness and backbone stretching, allowing us to obtain sensible curve fits. Thus, the DNA is represented

**Table 1 An overview of the three states $S_i = 0, 1, 2$ of link $i = 1, ..., N$ of our model DNA, with their interpretation, length measured in base pair distance in B-DNA, $l_0$, and free energy penalty $\Delta E$ discussed in the Methods section**

|  | State of DNA | Length | $\Delta E(S_i)$ |
|---|---|---|---|
| $S_i = 0$ | B-DNA | $l_0$ | 0 |
| $S_i = 1$ | Overstretched | $\gamma_1 l_0$ | $\varepsilon_1$ |
| $S_i = 2$ | Intercalated | $\gamma_2 l_0$ | $\varepsilon_2 - \mu$ |

**Table 2 Overview of energetic couplings $\Delta H(S_i, S_{i+1})$ between neighbouring segments depending on their state**

|  | $S_{i+1} = 0$ | $S_{i+1} = 1$ | $S_{i+1} = 2$ |
|---|---|---|---|
| $S_i = 0$ | 0 | $\lambda$ | 0 |
| $S_i = 1$ | $\lambda$ | 0 | $\eta$ |
| $S_i = 2$ | 0 | $\eta$ | $\delta$ |

The couplings $\lambda$, $\eta$ and $\delta$ are discussed in the Methods section

by a chain of $N$ segments, each of which corresponds to a single base pair. We first compute the fractions of the DNA molecule that occupy its various extensional states, and then we next use these fractions to include the effects of persistence and backbone stretching (see also Supplementary Note 3). The state of the $i^{th}$ base pair is given by a 'spin' $S_i$, which signifies either B-DNA (with length $l_0$, and spin $S_i = 0$), overstretched DNA ($\gamma_1 l_0$, $S_i = 1$), or intercalated DNA ($\gamma_2 l_0$, $S_i = 2$), where $\gamma_1$ and $\gamma_2$ are the elongation factors of the overstretched and intercalated states (see also Table 1). These elongation factors may be varied to reflect differences in the extension of the different states. The orientation of the $i^{th}$ base pair is given by the unit vector $\widehat{\mathbf{t}}_i$.

The free energy, $\varepsilon\big[\{\widehat{\mathbf{t}}_i\}, \{S_i\}\big]$, of a particular configuration of the chain, given by all unit vectors $\{\widehat{\mathbf{t}}_i\}$ and all spins $\{S_i\}$, is given, in units of thermal energy $k_B T$, by

$$\frac{\varepsilon\big[\{\widehat{\mathbf{t}}_i\}, \{S_i\}\big]}{k_B T} = \sum_{i=1}^{N}\left[ -\frac{f l_i}{k_B T}\widehat{\mathbf{t}}_i \cdot \widehat{\mathbf{z}} + \Delta E(S_i)\right] + \sum_{i=1}^{N-1} \Delta H(S_i, S_{i+1}). \tag{2}$$

The first term in the first line of Eq. (2) represents the work done by the entropic stretching force $f$ on the chain, applied in the $\widehat{z}$-direction. Here $l_i$ is the length of segment $i$. In the second term, $\Delta E(S_i) = \varepsilon_1 \delta_{S_i,1} + (\varepsilon_2 - \mu)\delta_{S_i,2}$, where $\varepsilon_1$ and $\varepsilon_2$ are the free energy costs, in units of thermal energy, of converting a single segment from B-DNA to overstretched and to intercalated DNA (see also Table 1).

The law of mass action enters through the (dimensionless) chemical potential of unbound intercalators, $\mu$, and follows directly from a grand-canonical description of the bound intercalators, in which the solution acts as a particle reservoir[38]. It ensures that more intercalators bind to the DNA if more are available in solution. The underlying assumption is that bound and unbound intercalators are in thermal and chemical equilibrium. We link the chemical potential $\mu$ directly to the overall molar intercalator concentration $C$ in the solution. For dilute solutions, we have $\mu = \ln(C/C_w)$, with $C_w$ the molar water concentration in the solution (55.6 M). Here, the reference chemical potential is absorbed in the binding free energy $\varepsilon_2$.

Interactions between neighbouring segments depend on their states, and are represented by the last term of Eq. (2). The relevant couplings that enter $\Delta H(S_i, S_{i+1})$ are given in Table 2. There are only three couplings of interest, denoted $\lambda$, $\eta$ and $\delta$, all in units of thermal energy. Others can be absorbed into renormalized values for $\lambda$, $\eta$ and $\delta$. $\lambda$ penalizes neighbouring B-DNA and overstretched base pair sequences, thus models the cooperative nature of the overstretching transition. $\delta$ penalizes two neighbouring intercalated segments, modelling next-neighbour exclusion of intercalators[12]. Finally, $\eta$ is a free energy penalty assigned to neighbouring overstretched and intercalated sequences, as suggested in our discussion of feature 2.

A summary of the states and the corresponding free energy penalties and cooperativities is given by Tables 1 and 2. This may seem like an excessive number of parameters. However, each of our parameters represent a physical effect in the interaction between DNA and intercalators, and each parameter controls a specific feature of the force-extension curves. In Supplementary Note 1, we describe how these parameters control the various features, as well as the fitting procedure we used to obtain the parameters.

Taking the Hamiltonian given in Eq. (2), we calculate the force-extension relation and the fraction of intercalated base pairs analytically from statistical mechanics. The partition function is given by

$$Z = \int_{\{\widehat{\mathbf{t}}_i\}} \sum_{\{S_i\}} \exp\left( -\frac{\varepsilon\big[\{\widehat{\mathbf{t}}_i\}, \{S_i\}\big]}{k_B T}\right), \tag{3}$$

which we evaluate in the long-chain limit $N \gg 1$ using the transfer matrix method[27]. We obtain the expectation value of the fraction of base pairs in the intercalated state by

$$\phi_2 = \langle \delta_{S_i,2}\rangle = \frac{\partial}{\partial \mu}\ln Z. \tag{4}$$

The fractions of base pairs in the B-DNA ($\phi_0$) and overstretched ($\phi_1$) states are given in Supplementary Note 2. Observing that in the absence of bend stiffness and backbone stretching we obtain the DNA extension as a superposition of three freely jointed chain contributions[39] representing B-DNA (Kuhn length $l_0$ with contribution $\phi_0$), overstretched DNA ($\gamma_1 l_0$, $\phi_1$) and intercalated DNA ($\gamma_2 l_0$, $\phi_2$) (Supplementary Note 3), we include the principle effects of persistence and backbone stretching by writing the force-extension relation as

$$\left\langle \frac{z}{L_0}\right\rangle = \phi_0 W_0(f, P_0, K_0) + \phi_1 W_1(f, P_1, K_1) + \phi_2 W_2(f, P_2, K_2), \tag{5}$$

where $L_0 = N l_0$ is the contour length of B-DNA, and $z/L_0 = 1$ corresponds to a DNA extension of $l_0$ per base pair. Here $\{P_0, P_1, P_2\}$ are the different persistence lengths, $\{K_0, K_1, K_2\}$ are the different stretching moduli and $\{W_0, W_1, W_2\}$ are the corresponding extensions of an extensible worm-like chain model of B-DNA, S-DNA and intercalated DNA, respectively, where we use the approximate force-extension relation found by Odijk[40]—details are supplied in Supplementary Note 3.

**Data availability**. Relevant data is available on request from the authors.

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

## Acknowledgements

We acknowledge support by the Netherlands Organization for Scientific Research (NWO) through project no. 712.012.007 (P.v.d.S.), the Human Frontier Science Program via a Research Grant (E.J.G.P.) and NWO VIDI (I.H.), VICI (G.J.L.W.) as well as an ERC Starting Grant (G.J.L.W.). We thank G. A. King for discussions.

## Author contributions

K.S. and A.S.B. are the primary authors of this work. P.v.d.S., C.S. and I.H. designed this study with the help of G.J.L.W. and E.J.G.P. A.S.B., B.t.B. and I.H. performed and analysed the experiments. K.S., M.S., P.v.d.S. and C.S. developed the theoretical framework. K.S., P.v.d.S., C.S., A.S.B. and I.H. wrote the article. All authors discussed the results and commented on the manuscript.

## Additional information

**Competing interests:** The authors declare no competing financial interests.

