## [Peer Review File · Nature Communications]

Reviewers' comments:

Reviewer #1 (Remarks to the Author):

This paper explains how the combination of intercalation and high force leads to the discovery of a hyperstretched DNA state. The paper is quite well written. There is a constant interplay between theory and experiment in this paper which is excellent. The most convincing evidence of this interplay is the match between figure 2b and figure 3e. I only have a few concerns that need attention.

1. On page 1 right column near the top the authors do not mention which intercalator extends DNA by 100%. Presumably, not all intercalators extend DNA by the same amount.
2. In the model the authors do not account for the elastic energy of the DNA in various phases. This stored elastic energy will be large because the forces applied are as large as 250pN. The authors do account for the stretching modulus in various phases in equation S5 of the supplement, so why do they stick with a freely jointed chain model in the 3-state model?
3. In page 7 the chemical potential μ appears. It is not clear to me why it enters ΔE as $\epsilon_2 - \mu$. I agree that it is supposed to account for mass-action, but how does a simple form $\epsilon_2 - \mu$ accomplish this?

Reviewer #2 (Remarks to the Author):

In this work the authors investigated the effects of intercalator (YOYO-1) on the B-to-S transition of DNA. They observed force-promoted intercalation of YOYO-1, YOYO-1 dependent increase in overstretching force, and disappearance of B-to-S transition plateau accompanied with appearance of a hyper-stretched DNA conformation bound with YOYO-1. They proposed a three-state model that can fully explain these observations.

Although the experimental results are very solid, most of the observed phenomena have been published in previous studies using different intercalators. The theoretical model is solid and provides nice insights into the mechanisms underlying the observed phenomena. The basis of their model is strongly backed up by fluorescence imaging of the mechanically stretched DNA. I also found the observed sequence-dependent YOYO-1 intercalation very interesting. Fig. 3 is the most interesting result to me.

In summary, I feel that this is a very interesting study with solid results. I would like to suggest the a minor revision before the publication of this work.

P2, lines 98-102, the authors mentioned, "Experiments were conducted in solution that favours DNA overstretching without the occurrence of single-stranded DNA sections originating from nicks and ends." They only cited the publication by Paik et al. that demonstrated that DNA overstretching transition does not need open ends of nick. The salt concentration and temperature dependent selection between strand-peeling and B-to-S transitions were extensively studied in the following papers that should also be cited:

Fu et al., (2010). "Two distinct overstretched DNA states." *Nucleic Acid Research* 38 :5594-5600.

Fu et al., (2011). "Transition dynamics and selection of the distinct S-DNA and strand unpeeling modes of double helix overstretching." *Nucleic Acids Research* 39:3473-3481.

Zhang et al., (2012). "Two distinct overstretched DNA structures revealed by single-molecule thermodynamics measurements." *Proceedings of the National Academy of Sciences*. 109:8103-8108.

Reviewers' comments:

Reviewer #1 (Remarks to the Author):

This paper explains how the combination of intercalation and high force leads to the discovery of a hyperstretched DNA state. The paper is quite well written. There is a constant interplay between theory and experiment in this paper which is excellent. The most convincing evidence of this interplay is the match between figure 2b and figure 3e. I only have a few concerns that need attention.

1. On page 1 right column near the top the authors do not mention which intercalator extends DNA by 100%. Presumably, not all intercalators extend DNA by the same amount.

We thank the reviewer for his/her remark and have clarified this in the revised manuscript. Specifically, we added the sentence (line 52ff): "Dense stacking interactions of the aromatic systems of typical planar mono-intercalators such as YO-PRO-1 and Sytox Orange and bis-intercalators such as YOYO-1 and POPO with neighboring base pairs result in an extension of DNA by 0.34 nm per intercalated moiety [Berman 1981, Ihmels 2005, Biebricher 2015]" and the references mentioned here.

As far as the model is concerned, we would like to stress that it has broader applicability, beyond simple intercalation that elongates DNA by 100%. Any interaction that results in a local elongation may be modeled by adjusting γ_2 , the value of the elongation factor of state 2. This will not change our qualitative findings, but will result in shifts of the various transition forces. We revised the Methods section to explicitly point out how the elongation factor may be varied in our model. (We added the sentence (line 542ff) "These elongation factors may be varied to reflect differences in the extension of the different states.")

2. In the model the authors do not account for the elastic energy of the DNA in various phases. This stored elastic energy will be large because the forces applied are as large as 250pN. The authors do account for the stretching modulus in various phases in equation S5 of the supplement, so why do they stick with a freely jointed chain model in the 3-state model?

Our choice to introduce our three-state model in its simplest, FJC form in the main text is for the sake of simplicity and in order to keep the number of free parameters to an absolute minimum. There is no qualitative difference in the transitions between stretching, overstretching and hyperstretching of FJCs vs other discrete DNA models such as the extensible Discrete Persistent Chain. The mathematics, however, is considerably more cumbersome and – we feel – does not add much new insight, where we note that the effects of backbone stretch and persistence have been characterized in detail before, affecting for instance the scaling of force with extension in the asymptotic regime. However, all fits and curves from our model as presented in the main text were obtained for the full model as put forth in the SI – with backbone extension and persistence accounted for by extensible WLCs. We now state this more clearly in the main text, and motivate our choice to focus initially on the three state FJC.

3. In page 7 the chemical potential μ appears. It is not clear to me why it enters ΔE as $\epsilon_2 - \mu$. I agree that it is supposed to account for mass-action, but how does a simple form $\epsilon_2 - \mu$ accomplish this?

*This expression arises straightforwardly from going from a canonical to a grand canonical ensemble in statistical mechanics. In our case the bulk solution with free intercalators acts as a reservoir that sets the chemical potential of the intercalators bound to the DNA. The latter constitutes the grand canonical system that we focus attention on. The chemical potential of the intercalators is known, provided the solution is dilute. See, for a derivation, e.g., D.J. Kraft W. K. Kegel and P. van der Schoot, *Biophysical Journal* 102 (2012), 2845–2855. Note in this context that the bound proteins follow a slightly more complicated version of Fermi-Dirac statistics, where each binding site constitutes a non-degenerate energy that can host at most one particle. We addressed this in the manuscript by adding a remark to the Methods section (line 561ff).*

Reviewer #2 (Remarks to the Author):

In this work the authors investigated the effects of intercalator (YOYO-1) on the B-to-S transition of DNA. They observed force-promoted intercalation of YOYO-1, YOYO-1 dependent increase in

overstretching force, and disappearance of B-to-S transition plateau accompanied with appearance of a hyper-stretched DNA conformation bound with YOYO-1. They proposed a three-state model that can fully explain these observations.

Although the experimental results are very solid, most of the observed phenomena have been published in previous studies using different intercalators. The theoretical model is solid and provides nice insights into the mechanisms underlying the observed phenomena. The basis of their model is strongly backed up by fluorescence imaging of the mechanically stretched DNA. I also found the observed sequence-dependent YOYO-1 intercalation very interesting. Fig. 3 is the most interesting result to me.

In summary, I feel that this is a very interesting study with solid results. I would like to suggest the a minor revision before the publication of this work.

Although a number of hints to the observed phenomena are indeed present in existing literature, we present here the first consistent and conclusive set of experiments and theoretical analysis that pins down and confirms the new hyperstretched state of DNA and its underlying physics. Our study has yielded additional new observations such as sequence-dependent de-intercalation and re-intercalation, quantitative insight into violation of neighbor exclusion of intercalated DNA, and, most importantly, a new comprehensive and predictive understanding of DNA mechanics under high load. We appreciate the positive recommendation of the reviewer and his/her suggestion for minor revisions that we address below.

P2, lines 98-102, the authors mentioned, "Experiments were conducted in solution that favors DNA overstretching without the occurrence of single-stranded DNA sections originating from nicks and ends." They only cited the publication by Paik et al. that demonstrated that DNA overstretching transition does not need open ends of nick. The salt concentration and temperature dependent selection between strand-peeling and B-to-S transitions were extensively studied in the following papers that should also be cited:

Fu et al., (2010). "Two distinct overstretched DNA states." *Nucleic Acid Research* 38 :5594-5600.

Fu et al., (2011). "Transition dynamics and selection of the distinct S-DNA and strand unpeeling modes of double helix overstretching." *Nucleic Acids Research* 39:3473-3481.

Zhang et al., (2012). "Two distinct overstretched DNA structures revealed by single-molecule thermodynamics measurements." *Proceedings of the National Academy of Sciences*. 109:8103-8108.

We thank the reviewer for pointing out these relevant references, and have added these to the manuscript bibliography.

Reviewers' Comments:

Reviewer #1 (Remarks to the Author):

The authors have addressed my concerns. The paper was already in good shape when it was first submitted, it's in better shape now. I recommend publication.

Reviewer #2 (Remarks to the Author):

The authors have revised their manuscript and addressed all my previous concerns.